# Treatment of Erythroid Precursor Cells from β-Thalassemia Patients with *Cinchona* Alkaloids: Induction of Fetal Hemoglobin Production

**DOI:** 10.3390/ijms222413433

**Published:** 2021-12-14

**Authors:** Cristina Zuccato, Lucia Carmela Cosenza, Matteo Zurlo, Ilaria Lampronti, Monica Borgatti, Chiara Scapoli, Roberto Gambari, Alessia Finotti

**Affiliations:** 1Section of Biochemistry and Molecular Biology, Department of Life Sciences and Biotechnology, University of Ferrara, 44121 Ferrara, Italy; cristina.zuccato@unife.it (C.Z.); luciacarmela.cosenza@unife.it (L.C.C.); matteo.zurlo@unife.it (M.Z.); ilaria.lampronti@unife.it (I.L.); monica.borgatti@unife.it (M.B.); 2Research Laboratory “Elio Zago” on the Pharmacologic and Pharmacogenomic Therapy of Thalassemia (Thal-LAB), University of Ferrara, 44121 Ferrara, Italy; 3Section of Biology and Evolution, Department of Life Sciences and Biotechnology, University of Ferrara, 44121 Ferrara, Italy; chiara.scapoli@unife.it; 4Interuniversity Consortium for Biotechnology (C.I.B.), 34148 Trieste, Italy

**Keywords:** β-thalassemia, fetal hemoglobin, γ-globin, HbF induction, K562 cells, *Cinchona* alkaloids, cinchonidine, quinidine, cinchonine, combined treatments

## Abstract

β-thalassemias are among the most common inherited hemoglobinopathies worldwide and are the result of autosomal mutations in the gene encoding β-globin, causing an absence or low-level production of adult hemoglobin (HbA). Induction of fetal hemoglobin (HbF) is considered to be of key importance for the development of therapeutic protocols for β-thalassemia and novel HbF inducers need to be proposed for pre-clinical development. The main purpose on this study was to analyze *Cinchona* alkaloids (cinchonidine, quinidine and cinchonine) as natural HbF-inducing agents in human erythroid cells. The analytical methods employed were Reverse Transcription quantitative real-time PCR (RT-qPCR) (for quantification of *γ-globin* mRNA) and High Performance Liquid Chromatography (HPLC) (for analysis of the hemoglobin pattern). After an initial analysis using the K562 cell line as an experimental model system, showing induction of hemoglobin and *γ-globin* mRNA, we verified whether the two more active compounds, cinchonidine and quinidine, were able to induce HbF in erythroid progenitor cells isolated from β-thalassemia patients. The data obtained demonstrate that cinchonidine and quinidine are potent inducers of *γ-globin* mRNA and HbF in erythroid progenitor cells isolated from nine β-thalassemia patients. In addition, both compounds were found to synergize with the HbF inducer sirolimus for maximal production of HbF. The data obtained strongly indicate that these compounds deserve consideration in the development of pre-clinical approaches for therapeutic protocols of β-thalassemia.

## 1. Introduction

β-thalassemias are among the most common inherited hemoglobinopathies worldwide, and are the result of more than 300 autosomal mutations of the gene encoding β-globin, causing an absence or low-level synthesis of this protein (and consequently of adult hemoglobin, HbA) in erythropoietic cells [1,2,3,4,5]. The phenotypes range widely from asymptomatic (β-thalassemia trait or carrier) to clinically relevant anemia, which is categorized as transfusion-dependent β-thalassemia (TDT, including thalassemia major) and non-transfusion-dependent β-thalassemia (NTDT, thalassemia intermedia) [1].

In the therapy of β-thalassemia, the induction of fetal hemoglobin (HbF) is considered to be of key importance for several concurrent reasons. In rare forms of β^0^-thalassemia, particularly those with large deletions responsible for δβ^0^-thalassemia or hereditary persistence of fetal hemoglobin (HPFH), high γ-globin chain production results in high levels of HbF, which is associated with a relatively benign phenotype [6,7,8,9]. More-recent clinical studies have disclosed that the naturally higher production of HbF improves the clinical course in a variety of patients with β-thalassemia [10,11,12,13,14]. Furthermore, approaches using genome editing are available in a form that is finalized to the induction of HbF following the elimination of genomic sequences encoding for *γ-globin* gene transcriptional repressors or genomic sequences targeted by these regulatory factors [15,16,17,18,19]. In this context, clinical trials are ongoing, such as NCT03655678 (A Safety and Efficacy Study Evaluating CTX001 in Subjects with Transfusion-Dependent β-Thalassemia), based on the use of autologous CRISPR-Cas9 Modified CD34^+^ Human Hematopoietic Stem and Progenitor Cells (hHSPCs) using CTX001 [20].

While some positive results have been described based on HbF induction by gene editing, the safety of this approach is still to be determined and, even in the case that its safety is demonstrated, the expected costs of this therapeutic approach would still be very high [21,22,23].

Accordingly, the validation of clinical relevance of already known HbF inducers and the characterization of novel HbF inducers are still projects of great interest [24,25,26,27].

In this context, a recent paper was published on the effects of *Cinchona* alkaloids (cinchonidine and quinidine) as natural fetal hemoglobin-inducing agents, using the human erythroleukemia K562 cell line as in vitro experimental model system [28]. In this study, *Cinchona* alkaloids showed dose-dependent induction of erythroid differentiation, increased production of HbF and high contents of *γ-globin* mRNA. Despite the fact that this study was based only on the use of K562 cells as a model system, it demonstrated that *Cinchona* alkaloids should be considered for the development of therapeutic protocols for β-thalassemia [28]. In addition, it should be underlined that molecules belonging to this family have been extensively used in therapy for other indications than β-thalassemia and, therefore, might be considered as “repurposed drugs” with a facilitated strategy to reach technology transfer [29,30,31,32,33,34]. For instance, quinine, quinidine, cinchonidine and cinchonine alkaloids had a powerful bioimpact as anti-malarial drugs [29,30,31]. Quinidine is also used to treat cardiac arrhythmias because it inhibits fibrillation [32,33,34].

Three are the main objectives of our study: (a) to confirm the data already published on the effects of *Cinchona* alkaloids (cinchonidine, quinidine and cinchonine) on K562 erythroleukemia cells; (b) to verify whether these molecules might potentiate the activity of other HbF inducers; (c) to verify whether the more promising *Cinchona* alkaloids tested induce HbF in erythroid progenitor cells isolated from β-thalassemia patients.

In terms of combined treatments, we decided to employ sirolimus (SIR, rapamycin) [35] in the co-treatment experiments for the following reasons: (a) sirolimus increases HbF in cultures from β-thalassemia patients with different basal HbF levels [36,37,38,39]; (b) sirolimus increases the overall Hb content per cell [37]; (c) sirolimus selectively induces *γ-globin* mRNA accumulation, with only minor effects on β-globin and *α-globin* mRNAs [36,37]; (d) sirolimus was found to induce HbF in vivo in mouse model systems [40,41,42]; and, more importantly, (e) sirolimus was able to increase HbF in sickle-cell disease (SCD) patients [43,44]. For these reasons, sirolimus obtained the Orphan Drug Designation by the European Medicinal Agency (EMA, Europe) and by the Food and Drug Administration (FDA, USA) for both β-thalassemia and SCD. Of relevance for this study, two ongoing clinical trials are based on sirolimus: NCT03877809 (A Personalized Medicine Approach for β-thalassemia Transfusion Dependent Patients: Testing sirolimus in a First Pilot Clinical Trial) and NCT04247750 (Treatment of β-thalassemia Patients with Rapamycin: From Pre-clinical Research to a Clinical Trial) [45]. Finally, it should be noted that sirolimus is a well-known drug, since it is employed for other therapeutic indications, such as kidney transplantation [46,47], cardiac [48] and liver [49] transplantation, lupus erythematosus (SLE) [50], lymphangioleiomyomatosis (LAM) [51], tuberous sclerosis complex [52] and different types of cancers [53,54,55].

## 2. Results

### 2.1. Cinchonidine, Quinidine and Cinchonine Induce Differentiation of K562 Erythroleukemia Cells

Figure 1 shows that cinchonidine (CincD), quinidine (QuinD) and cinchonine (CincN) are all able to induce erythroid differentiation of human K562 cells in a concentration-dependent fashion. Determinations were performed after 5, 6 and 7 days of differentiation induction.

After this induction period, cells were stained with benzidine in order to identify the hemoglobin-containing cells [26,36,37,38]. These data confirm the already-reported effects of these *Cinchona* alkaloids on the K562 cell system [28]. The induction of K562 differentiation was found to be associated, as expected, with the inhibition of cell proliferation, as reported in the K562 cell system for several other HbF inducers, such as hydroxyurea, mithramycin, rapamycin, cisplatin analogues and trimethylangelicin [25,26,27,37,56,57,58,59]. This effect of cinchonidine, quinidine and cinchonine on K562 cell proliferation is presented and comparatively analyzed in Appendix A. The induction of K562 erythroid differentiation by the studied *Cinchona* alkaloids was similar (with respect to the extent of the induced proportion of benzidine-positive cells) to that of sirolimus (rapamycin), despite the fact that the effects of sirolimus were appreciable at 100–200 nM (Appendix A). These differences among putative erythroid inducers were fully expected [60]. For example, butyrates are active at mM concentrations [61].

### 2.2. Cinchonidine, Quinidine and Cinchonine Potentiate Sirolimus-Induced Differentiation of K562 Erythroleukemia Cells

In order to verify possible combined effects of *Cinchona* alkaloids and sirolimus, we treated K562 cells simultaneously with different concentrations of these molecules. The obtained results of the treatment, reported in Figure 2, show that CincD, QuinD and CincN were able to further increase the erythroid differentiation activity of sirolimus, when this HbF inducer was used at 100 nM (Figure 2, panels A, C and E) and 200 nM (Figure 2, panels B, D and F).

In this experiment, K562 cells were cultured with increasing concentrations of cinchonidine, quinidine and cinchonine in the presence of 100 nM and 200 nM sirolimus, as indicated. The proportion of benzidine-positive K562 cells was determined after 5, 6 and 7 days of treatment. The data presented in Figure 2G are related to the use of sub-optimal concentrations of CincD, QuinD and CincN (60, 40 and 50 µM, respectively) in the presence of 100 and 200 nM sirolimus. When the inducers (either sirolimus, or cinchonidine, quinidine and cinchonine) were used alone, the proportion of benzidine-positive hemoglobin-containing K562 cells was about 10–30%. On the contrary, when the inducers were used in combination, the proportion of benzidine-positive hemoglobin-containing K562 cells was always found to exceed 50% (the maximum level was reached using cinchonidine in combination with 200 nM sirolimus). The increased % of benzidine-positive cells compared with single administrations was always found to be statistically highly significant (*p* < 0.001). Untreated K562 displayed a proportion of benzidine-positive cells which never exceeded 2–5% (see, for instance, the microphotographs presented in the left part of Figure 3).

Figure 3 shows a representative microscopic analysis of the benzidine assays performed under some of the different experimental conditions described in Figure 2. The data give clear evidence that, in addition to the already reported increase in the proportion of benzidine-positive cells, and additional feature of combined treatments was evident, i.e., that all the benzidine-positive cells were brightly stained with benzidine-hydrogen peroxide solution (black arrowheads).

On the contrary, the presence of slightly stained benzidine-positive cells was clearly appreciable in singularly treated K562 cells (white arrowheads). While the representative data shown in Figure 3 were obtained using CincD plus SIR combination, the data obtained using QuinD/SIR and CincN/SIR combinations were found to be very similar (as shown in Appendix A). Although this assay was not quantitative, the results obtained sustain the concept that the combined treatments lead to the highest levels of increase in the proportion of brightly stained benzidine-positive K562 cells. In order to verify this hypothesis using a more quantitative assay, RT-qPCR was performed on isolated RNA.

### 2.3. The Effects of Cinchonidine, Quinidine and Cinchonine on K562 Erythroid Differentiation Are Associated with a Modulation of Expression of α-Globin and γ-Globin Genes

Figure 4 shows that the erythroid differentiation induced by cinchonidine, quinidine and cinchonine was associated with an increase in the production of *α-globin* (Figure 4A,C,E) and *γ-globin* mRNAs (Figure 4B,D,F). The analysis of the expression of *α-globin* and *γ-globin* mRNAs was performed by RT-qPCR.

Interestingly, when used singularly, the increases in the production of both mRNAs induced by CincD (Figure 4A,B), QuinD (Figure 4C,D), and CincN (Figure 4E,F) were found to be similar to those found when 100 nM and 200 nM sirolimus was employed (*p* > 0.05). In addition, the increased levels of *γ-globin* mRNAs were similar to those originally described by Iftikhar et al. [28]. On the contrary, when cinchonidine, quinidine and cinchonine were used in combination with 100 nM and 200 nM sirolimus, a sharp increase in the content of *α-globin* mRNA (*p* < 0.01) and a less extensive but still significant (*p* < 0.05) increase in *γ-globin* mRNAs were found, fully in agreement with the effects of these compounds on sirolimus-induced K562 erythroid differentiation (Figure 2 and Figure 3). In Figure 4, relevant examples of the p values obtained are shown, while the complete statistical analysis is presented in Appendix A.

The effects of CincD, QuinD, and CincN were dose-dependent, but clearly evident even when sub-optimal concentrations of compounds were employed.

Among the different combinations studied, those based on cinchonidine and quinidine were found to be the most effective in terms of inducing increases in the proportion of benzidine-positive cells and increased expression of globin genes (Appendix A). Therefore, these two compounds were selected for further studies using erythroid precursor cells (ErPCs) from β-thalassemia patients as an experimental model system.

### 2.4. Cinchonidine and Quinidine Induce HbF and γ-Globin mRNA in Erythroid Precursor Cells (ErPCs) from β-Thalassemia Patients

Patients were recruited at the Thalassemia Centre of Azienda Ospedaliera-Universitaria S. Anna (Ferrara, Italy). In total, 10 patients were enrolled. Informed written consent from all participants was obtained before recruiting them into the study. Different genotypes were present in the recruited cohort: five patients were β^0^39/β^+^IVSI-110, two patients were β^+^IVSI-110/β^+^IVSI-110 and three patients were β^0^39/β^0^39. ErPCs were isolated from the β-thalassemia patients and cultured, as described elsewhere [59], with erythropoietin in the presence of sirolimus, cinchonidine and quinidine administered alone (this was performed in ErPC cultures from all the ten patients) or in combination (this was performed in ErPC cultures from five patients).

In Figure 5, the HPLC profiles of two representative ErPC populations are shown, one isolated from patient #10 (Figure 5A–C) and the other from patient #5 (Figure 5D–F), exhibiting a differential response to cinchonidine (Figure 5B,E) and quinidine (Figure 5C,F) treatment.

The ErPCs from patient #10 exhibited a 69.57% and 118.04% increase in HbF after treatment with cinchonidine and quinidine, respectively (see the raw data shown in Appendix A). A lower increase in HbF was obtained when the ErPCs from patient #1 were employed (in this case, the HbF increase was 5.40% and 9.53% for cinchonidine and quinidine, respectively).

Figure 6 shows the effects of cinchonidine and quinidine on the ErPCs from all the 10 recruited β-thalassemia patients. All the raw data are reported in Appendix A. As clearly evident, increases in the proportion of HbF (% of all the accumulated hemoglobins) were found in the treated ErPCs from most of the recruited β-thalassemia patients (Figure 6A).

Figure 6B presented this increase in the % of HbF with respect to untreated ErPCs and gives evidence for an HbF increase comparable to that of sirolimus and the established HbF inducer hydroxyurea.

In fact, the HbF increase was significant in the treated cells when the obtained values were compared to those found in untreated cells (Figure 6B).

As expected for HbF inducers, the differences were not statistically significant when the data of CincD- and/or QuinD-treated ErPCs were compared with those obtained using SIR or HU.

The HbF increase was, as expected, associated with increase in *γ-globin* mRNA (Figure 6C). Moreover, in this case, the increases in *γ-globin* mRNA (2.26 ± 0.37 and 2.04 ± 0.36 folds for cinchonidine and quinidine, respectively) were similar to the increases found in sirolimus-treated (2.21 ± 0.29) and hydroxyurea-treated (2.41 ± 0.35) ErPCs. The expected slight inhibitory effects of CincD and QuinD on ErPC cell proliferation were similar to those of the validated HbF inducers SIR and HU (Appendix A).

### 2.5. Cinchonidine and Quinidine Potentiate Sirolimus-Mediated Induction of HbF and γ-Globin mRNA in ErPCs from β-Thalassemia Patients

Figure 7 shows the data obtained when ErPCs from five patients were treated, in addition to the treatments already mentioned in Section 2.4, with cinchonidine and quinidine in the presence of 100 nM sirolimus. The data reported are related to increase in the % of HbF and of *γ-globin* mRNA content. As clearly evident, the ErPC cultures exhibiting the highest levels of % of HbF and increased *γ-globin* mRNA are those treated with the two alkaloids and sirolimus (the raw data are reported in Appendix A). For instance, the % increase in HbF was 66.30 ± 25.09 and 82.30 ± 37.18 in ErPCs co-treated with sirolimus plus cinchonidine or sirolimus plus quinidine, respectively. These values were higher than those found when single drugs were added (20.45 ± 13.82, 31.61 ± 11.62 and 54.83 ± 19.52 for sirolimus, cinchonidine and quinidine, respectively). When the values relative to the treatments with CincD plus SIR and with QuinD plus SIR were compared to the treatment with the reference HbF inducers HU and SIR, the differences in the % of HbF incresae were found to be highly significant (Figure 7A, upper part of the panel). On the contrary, when the values of CincD or QuinD are compared to HU or SIR, the differences were not significant (*p* > 0.2).

### 2.6. Treatment of ErPCs from β-Thalassemia Patients with Cinchonidine and Quinidine Is Associated with a Sharp Decrease in the Free α-Globin Chains

Reduction in the excess α-globin should be considered as an important objective in the development of therapeutic interventions of β-thalassemia, since the excess α-globin decreases the lifespan of the red-blood cells, causes ineffective erythropoiesis and is a major determinant of the clinical severity of β-thalassemia [42].

The effects of cinchonidine and quinidine on the free α-globin chains produced by the ErPCs from the recruited β-thalassemia patients displaying α-globin > 2.5 are summarized in Figure 8. All the raw data are reported in Appendix A.

As clearly evident, a decrease in the % of the free α-globin peak was found in the treated ErPCs from most of the recruited β-thalassemia patients (see Appendix A). Of great interest is a lower reduction in the free α-globin chains when ErPCs were treated with sirolimus or hydroxyurea. An average α-globin peak reduction of 27.98% was found with sirolimus, while the % average reduction with hydroxyurea was only 10.78%. Higher and more significant reductions (*p* = 0.024504 and 0.006129, respectively) were found with cinchonidine and quinidine (49.15% and 58.79%, respectively). These data suggest that, in addition to increased expression of γ-globin genes and HbF production, cinchonidine and quinidine might exert their beneficial effects on ErPCs through a decrease in the excess free α-globin chains.

## 3. Discussion

Induction of fetal hemoglobin (HbF) is considered a very promising strategy in the therapy of β-thalassemia and sickle-cell disease (SCD). The gene-therapy-mediated induction of γ-globin gene expression and HbF production in erythroid cells was described. In addition, approaches using genome editing are available in forms that are finalized to the induction of HbF following the elimination of genomic sequences encoding for transcriptional repressors or genomic sequences targeted by these regulatory factors [15,16,17,18,19].

While some positive results have been described based on HbF induction by gene therapy and gene editing, the safety of these approaches are still to be determined. In addition, it is expected that the costs of these therapeutic interventions will be very high [21,22,23].

Accordingly, the validation of the clinical relevance of already-known HbF inducers and the characterization of novel HbF inducers are still projects of great interest [24,25,26,27]. In this context, a paper was recently published by Iftikhar et al. on the effects of *Cinchona* alkaloids (cinchonidine and quinidine) as natural fetal hemoglobin-inducing agents in human erythroleukemia cells [28]. This study was very interesting, despite the fact that the key results were obtained using only K562 cells as an in vitro experimental model system.

The key conclusions of our study are the following: (a) cinchonidine and quinidine are inducers of an increase in the % of HbF in erythroid progenitor cells isolated from β-thalassemia patients; (b) cinchonidine and quinidine potentiate the activity of sirolimus (a HbF inducer employed in clinical trials). These data sustain the concept that cinchonidine and quinidine should be considered for further studies aimed at developing protocols for the treatment of β-thalassemia patients. Our data show that the HbF induction efficiency of cinchonidine and quinidine is similar to that of hydroxyurea (the reference HbF inducer) and sirolimus (Figure 6).

In addition to the induction of changes in HbF expression, cinchonidine and quinidine might also act through a reduction in the excess free α-globins present in the erythroid cells of β-thalassemia patients. This reduction should be considered a key objective in the use of molecules for therapeutic interventions in the management of β-thalassemia, since the excess α-globin is one of key factors causing short lifespans of the red-blood cells with associated ineffective erythropoiesis [42]. Interestingly, this therapeutic relevant target is reached very efficiently using cinchonidine and quinidine, as both are more efficient in reducing the excess free α-globin than hydroxyurea and sirolimus (Figure 8). Further studies will clarify whether the reduction in the excess α-globin is associated with the activation of autophagy, as proposed elsewhere [42].

In terms of combined treatments, we decided to employ sirolimus [35] as this HbF inducer is, at present, employed in two ongoing clinical trials: NCT03877809 (A Personalized Medicine Approach for β-thalassemia Transfusion Dependent Patients: Testing sirolimus in a First Pilot Clinical Trial) and NCT04247750 (Treatment of β-thalassemia Patients with Rapamycin: From Pre-clinical Research to a Clinical Trial) [45]. Further studies will verify whether the *Cinchona* alkaloids employed in this study potentiate the activity of other HbF inducers, including hydroxyurea, that are extensively employed in the treatment of β-thalassemia and sickle-cell disease [62,63]. One of the limits of our study is that the mechanism(s) of action was not experimentally evaluated. Further studies are required to understand this specific issue in our ErPC model system. However, published studies support the hypothesis that the mechanism(s) of action of *Cinchona* alkaloids and sirolimus are sharply different. In fact, *Cinchona* alkaloids are reported to inhibit cytochrome P450 enzyme 2D6 and the transport protein P-glycoprotein [64,65], while sirolimus is firmly established as an mTOR inhibitor [35,66].

The data obtained in our study strongly support the concept that cinchonidine and quinidine might be employed in combination with sirolimus in order to maximize its effects on in vivo-treated β-thalassemia patients.

In this respect, it is interesting to observe that cinchonidine and quinidine might be more active than hydroxyurea, which is one of the most important reference compounds when clinical treatment of β-thalassemia and sickle-cell disease is considered [62,63]. Further studies employing analyses of the effects on transcriptome and proteome, as well as confirming the presence of increased HbF within selected cell populations, are needed in order to verify whether the increase in the % of HbF reported in the present study is accompanied by a clinically relevant increase in the content of HbF in each treated erythroid cell.

Moreover, in order to propose a possible protocol for therapeutic purposes, a proof-of-principle showing in vivo effects on animal model systems is highly recommended, as well as a careful analysis of the relationship between the effects on HbF and the presence of DNA polymorphisms associated with the predisposition of patients to high HbF induction.

## 4. Materials and Methods

### 4.1. Patients Recruitment

Cultures of erythroid progenitors were derived from the peripheral blood of β-thalassemia patients. Patients were recruited at the Day Hospital Thalassemia and Hemoglobinophaties of Azienda Ospedaliera-Universitaria S. Anna (Ferrara, Italy). All the patients received a patient information sheet to read and time to clarify doubts with investigators before consenting. All the participants signed an informed consent form on the basis of approvals of the Ethical Committee in charge of human studies at the University Hospital. The recruited patients were all transfusion dependent and not under hydroxyurea therapy. Treatments were performed on cultured ErPCs derived from patients blood isolated just before transfusion.

### 4.2. Chemical Reagents for Cell Culture Treatments

The reagents used for K562 treatments (rapamycin (sirolimus, SIR), cat. R0395; hydroxyurea (HU), cat. H8627; cinchonine (CincN), cat. 27370; cinchonidine (CincD), cat. C80407; quinidine (QuinD), cat. 22600) were purchased from Sigma Aldrich (St. Louis, MO, USA). HU was solubilized in sterile deionized H_2_O, whereas rapamycin, cinchonine, cinchonidine and quinidine were solubilized in ethanol and stored at −20 °C. Stock solution of rapamycin was prepared at 5 mM and 20 mM for each of the *Cinchona* alkaloids used. We used a concentration of sirolimus known to induce both K562 and erythroid precursor cells from β-thalassemia patients [37]. These stocks were further diluted to the indicated concentrations in culture medium prior to experimentation. All the treatments were performed by adding the compounds once at the beginning of the culturing period.

### 4.3. Human K562 Cell Cultures

The human leukemia K562 [26,36] cells were cultured in a humidified atmosphere of 5% CO_2_/air in RPMI 1640 medium (RPMI 1640 medium; Lonza, Verviers, Belgium) supplemented with 10% (vol/vol) fetal bovine serum (FBS; Biowest, Nuaille, France) and 1% penicillin–streptomycin (Euroclone, Milano, Italy). Cell growth was studied by determining the cell number per ml with a Z2 Coulter Counter (Beckman Coulter, Fullerton, CA, USA).

### 4.4. In Vitro Culture of Erythroid Progenitors from β-Thalassemia Patients

The two-phase liquid culture procedure was employed as previously described [37,59]. Mononuclear cells were isolated from peripheral blood samples of β-thalassemia patients: 20–25 mL of peripheral blood were collected before transfusion from patients who gave informed consent. A mixture of blood and PBS 1× at a 1:1 ratio was stratified on top of Lympholyte^®^-H Cell Separation Media (Cedarlane, Burlington, NC, USA). After isolation, the mononuclear cell layer was washed three times by adding 1× PBS solution and seeded in α-minimal essential medium (α-MEM; Sigma Aldrich, St. Louis, MO, USA) supplemented with 10% FBS (FBS; Biowest, Nuaille, France), 1 µg/mL cyclosporine A (Sigma Aldrich, St. Louis, MO, USA), 10% conditioned medium from the 5637 bladder carcinoma cell line culture and stem cell factor (SCF, Life Technologies, Monza, MB, Italy) at the final concentration of 10 ng/ml. The cultures were incubated at 37 °C, under an atmosphere of 5% CO_2_, with extra humidity. After 7 days in this phase-I culture, the non-adherent cells were harvested from the flask, washed in 1x PBS, and then cultured in phase-II medium, composed of α-MEM medium (Sigma Aldrich, St. Louis, MO, USA), 30% FBS (FBS; Biowest, Nuaille, France), 1% deionized bovine serum albumin (BSA, Sigma Aldrich, St. Louis, MO, USA), 10^−5^ M β-mercaptoethanol (Sigma Aldrich, St. Louis, MO, USA), 2 mM L-glutamine (Sigma Aldrich, St. Louis, MO, USA), 10^−6^ M dexamethasone (Dexamethasone 21-phosphate disodium salt; Sigma Aldrich, St. Louis, MO, USA) and 1 U/mL human recombinant erythropoietin (EPO) (Tebu-bio, Magenta, Milano, Italy) and stem cell factor (SCF) at the final concentration of 10 ng/mL. Erythroid precursor cells’ differentiation was assessed by benzidine staining [59].

### 4.5. Reverse Transcription and Quantitative Real-Time PCR (RT-qPCR)

For gene expression analysis, 500 ng of total RNA were reverse transcribed using the TaqMan^®^ Reverse Transcription Reagents kit and random hexamers (Applied Biosystems, Foster City, CA, USA). The RT-qPCR assay was carried out using gene-specific double fluorescently labeled probes. The reaction mixture had a final volume of 25 μL and was composed of Prime Time^®^ Gene Expression Master Mix 1× (IDT, Tema Research, Castenaso, BO, Italy), the pairs of forward and reverse primers (α, β, γ, together or *GAPDH*, *RPL13A*, *ACTB* together) used at 500 nM concentration and the probes (α, β, γ, together or *GAPDH*, *RPL13A*, *ACTB* together) used at 250 nM concentration. The probes that contained 6-carboxyfluorescein (FAM) and hexachloro-6-carboxyfluorescein (HEX) as chromogenic molecules at 5′ were quenched by the Iowa Black^®^ FQ molecule at 3′, while probes that contained indocarbocyanine (Cy5) were quenched by Iowa Black^®^ RQ. After an initial step for the denaturation at 95 °C for 2 min, the reactions were performed for 50 cycles consisting of two phases, 95 °C for 10 s and 60 °C for 45 s.

### 4.6. HPLC Analysis of Hemoglobins

To evaluate the effective quantity of the various types of hemoglobin produced by the cultured erythroid cells after treatment, High-Performance Liquid Chromatography was performed. The ErPCs were centrifuged at 2000 rpm for 6 minutes and washed with PBS (Phosphate buffered saline). The pellet was then resuspended in a predefined volume of water for HPLC (Sigma-Aldrich, St. Louis, MO, USA). This was followed by 3 freeze/thaw cycles on dry ice in order to lyse the cells and obtain the protein extracts. Lysates were centrifuged for 5 min at 14,000 rpm and the supernatant was collected. Hemoglobin analysis was performed by loading the protein extracts into a PolyCAT-A cation exchange column, and they were then eluted in a sodium-chloride-BisTris-KCN aqueous mobile phase using the HPLC Beckman Coulter Instrument System Gold 126 Solvent Module-166 Detector, which allowed us to obtain a quantification of the hemoglobins present in the sample. The reading was performed at a wavelength of 415nm, and a commercial solution of purified human HbAF (Sigma-Aldrich) extracts was used as a standard. The values thus obtained were processed using “32 Karat software”.

### 4.7. Statistical Analysis

All the data were normally distributed and presented as mean ± S.E.M. Statistical differences between groups were compared using a paired *t*-test or a one-way repeated measures ANOVA (ANalyses of VAriance between groups) followed by LSD post-hoc tests. Statistical differences were considered significant when *p* < 0.05 (*) and highly significant when *p* < 0.01 (**).

## Figures and Tables

**Figure 1 ijms-22-13433-f001:**
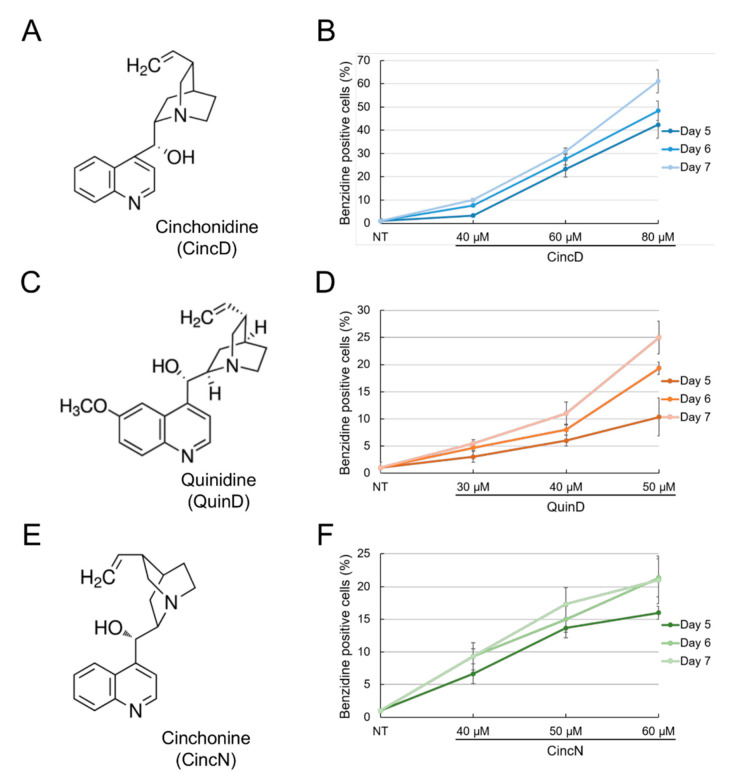
Effects of cinchonidine, quinidine and cinchonine on erythroid differentiation of K562 cells. (**A**,**C**,**E**) Structure of cinchonidine (CincD) (**A**), quinidine (QuinD) (**C**) and cinchonine (CincN) (**E**). (**B**,**D**,**F**) Effects on erythroid differentiation of K562 cells. K562 cells were cultured in the presence of the indicated concentrations of cinchonidine (**B**), quinidine (**D**) and cinchonine (**F**) and analysis of the proportion of benzidine-positive hemoglobin-containing cells was performed after 5, 6 and 7 days of cell culture. The data represent the average ± S.E.M. (*n* = 3).

**Figure 2 ijms-22-13433-f002:**
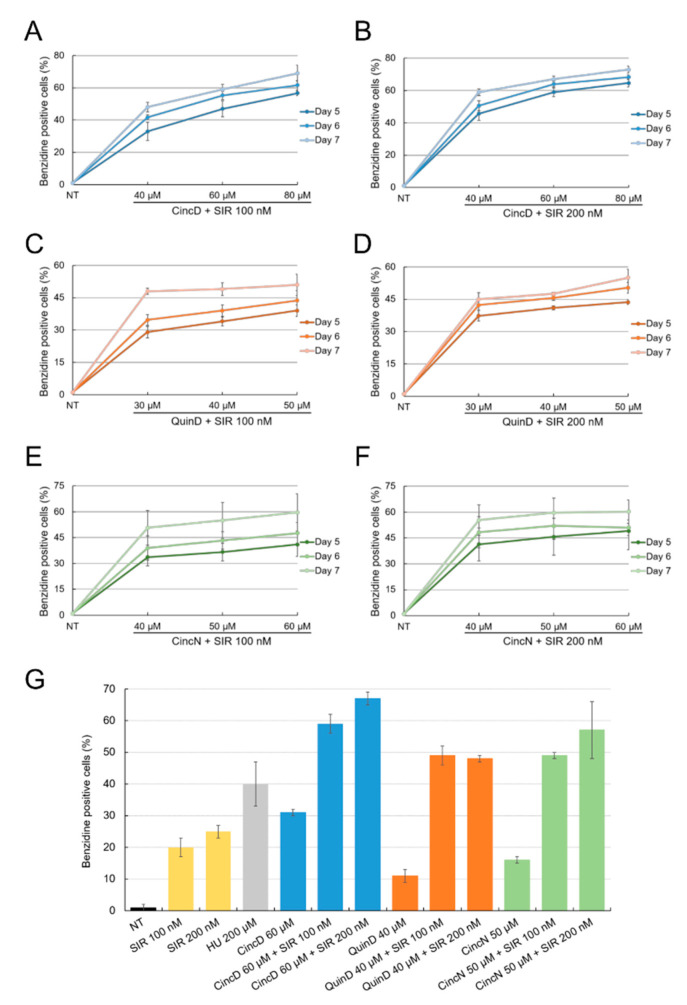
Effects of cinchonidine, quinidine and cinchonine on K562 erythroid differentiation induced by sirolimus. K562 cells were cultured with increasing concentrations of CincD (**A**,**B**), QuinD (**C**,**D**) and CincN (**E**,**F**) in the presence of 100 nM (**A**,**C**,**E**) and 200 nM (**B**,**D**,**F**) sirolimus (SIR). The proportion of benzidine-positive K562 cells was determined after 5, 6 and 7 days of treatment, as indicated. (**G**) Comparative analysis of the data obtained at day 7 of treatment. The data represent the average ± S.E.M. (*n* = 3).

**Figure 3 ijms-22-13433-f003:**
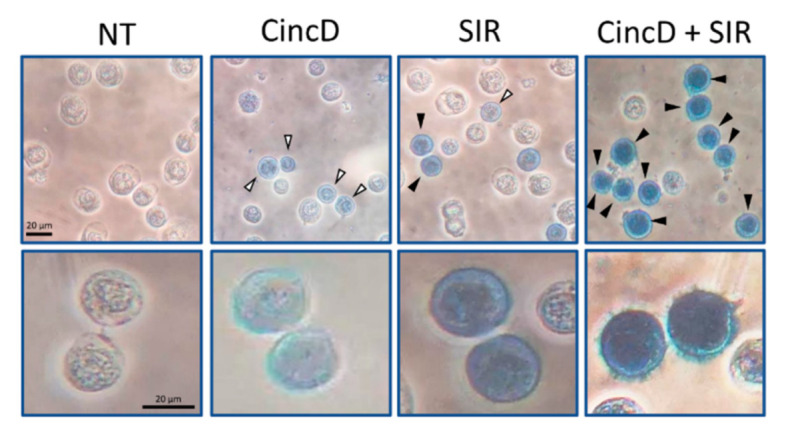
Benzidine-positive cells in untreated K562 cell cultures (NT) or in K562 cell cultures treated with CincD, SIR and CincD + SIR, as indicated. White arrowheads = slightly benzidine-stained cells; black arrowheads = brightly benzidine-stained cells. Magnitude: 20× (upper panels) and 40× (lower panels). Scale bar, 20 μm.

**Figure 4 ijms-22-13433-f004:**
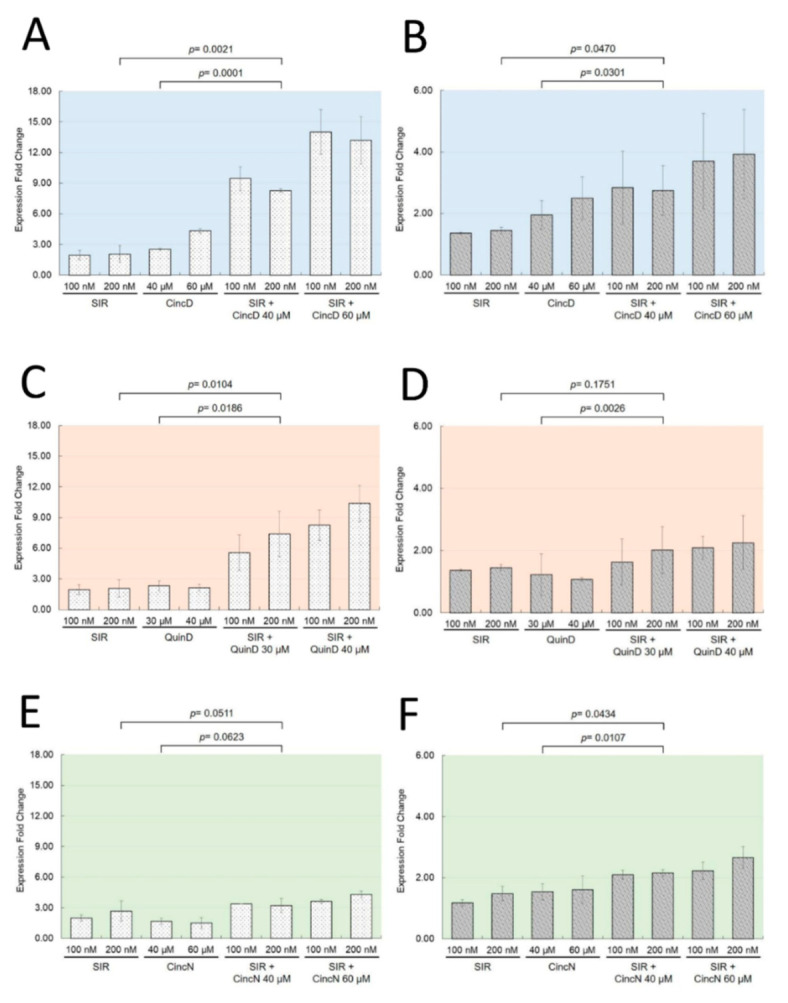
Effects of cinchonidine, quinidine and cinchonine on *α-globin* (left) and *γ-globin* (right) mRNAs. K562 cells were cultured with increasing concentrations of CincD (**A**,**B**), QuinD (**C**,**D**) and CincN (**E**,**F**) in the presence of 100 nM and 200 nM sirolimus (SIR) as indicated. After 5 days of treatment, RNA was isolated and RT-qPCR was performed using primers amplifying *α-globin* (**A**,**C**,**E**) and *γ-globin* (**B**,**D**,**F**) sequences. The increase in expression of *α-globin* and *γ-globin* mRNAs is presented as fold change with respect to control untreated cells. The data represent the average ± S.E.M. (*n* = 3).

**Figure 5 ijms-22-13433-f005:**
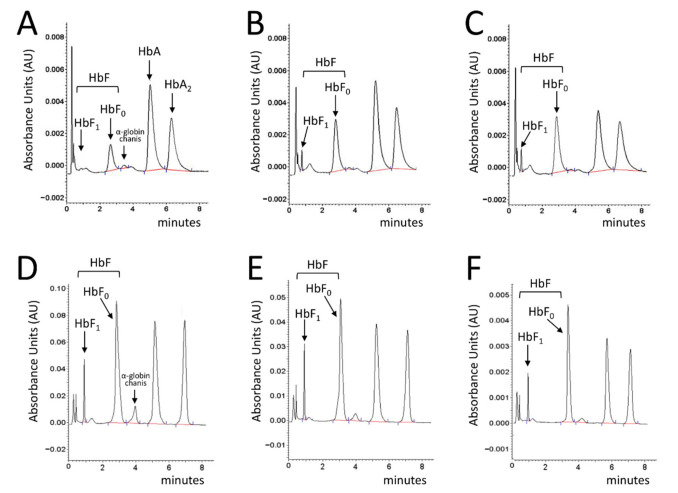
HPLC profile of cytoplasm isolated from ErPCs treated with cinchonidine and quinidine. ErPCs isolated from patient #10 (**A**–**C**) and patient #1 (**D**–**F**) were treated with 60 µM cinchonidine (**B**,**E**) and 30 µM quinidine (**C**,**F**) for 7 days and the lysate was analyzed by HPLC. HPLC profiles of untreated ErPCs are depicted in panels (**A**,**D**). The peaks corresponding to HbF, free α-globin chains, HbA and HbA_2_ are shown.

**Figure 6 ijms-22-13433-f006:**
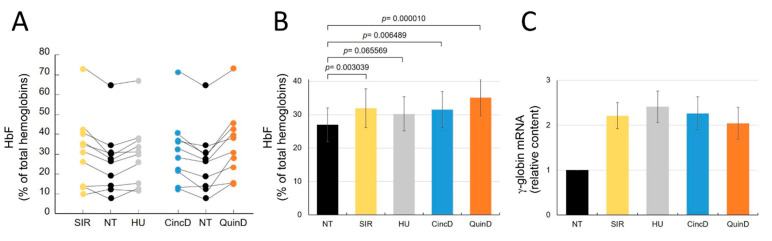
Effects of cinchonidine and quinidine on the HbF production and *γ-globin* gene expression in treated ErPCs. ErPcs isolated from 10 β-thalassemia patients were treated with 100 nM sirolimus (SIR), 100 µM hydroxyurea (HU), 60 µM cinchonidine (CincD) and 30 µM quinidine (QuinD) for 7 days. The lysate was analyzed by HPLC for HbF quantification (% with repect to total HbF produced) and the RNA was analyzed by RT-qPCR for determination of the relative *γ-globin* mRNA content. The raw data of HbF content are shown in the Appendix A. The % of HbF following the different treatments are reported in (**A**,**B**). The increase in *γ-globin* mRNA content is shown in (**C**). The results of panels (**B**,**C**) are reported as mean ± S.E.M. (*n* = 10).

**Figure 7 ijms-22-13433-f007:**
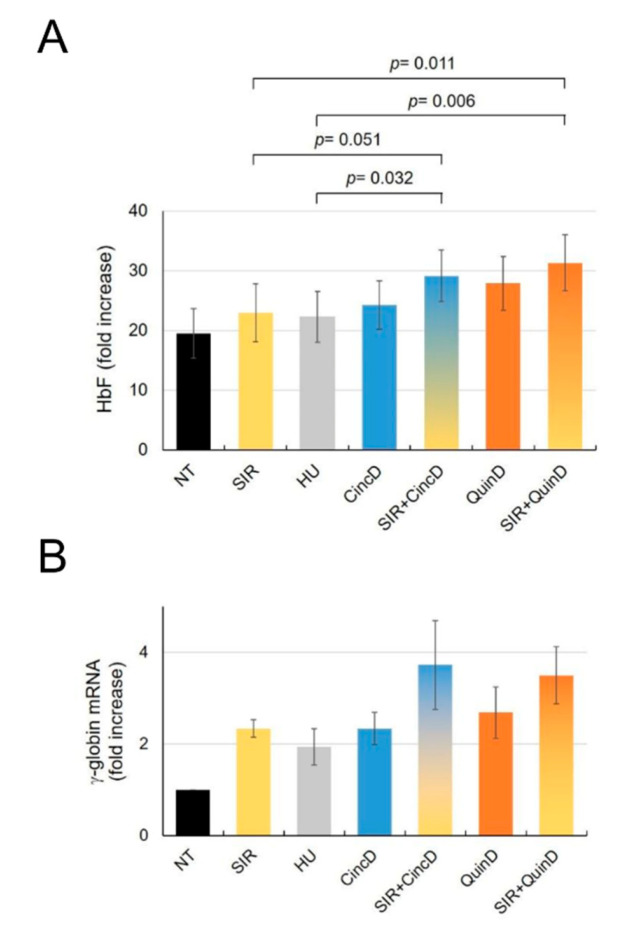
Effects of co-treatment of ErPCs with sirolimus and cinchonidine or quinidine. ErPCs isolated from five β-thalassemia patients were treated, as indicated, for 7 days with 100 nM sirolimus (SIR), 100 μM hydroxyurea (HU), 60 µM cinchonidine (CincD), 30 µM quinidine (QuinD) or with SIR + CincD or SIR + QuinD. The lysates were analyzed by HPLC for HbF quantification and the RNA by RT-qPCR for determination of the *γ-globin* mRNA content. The average increases in HbF are reported in (**A**), the increase in *γ-globin* mRNA content is shown in (**B**). The data represent the mean ± S.E.M. (*n* = 5).

**Figure 8 ijms-22-13433-f008:**
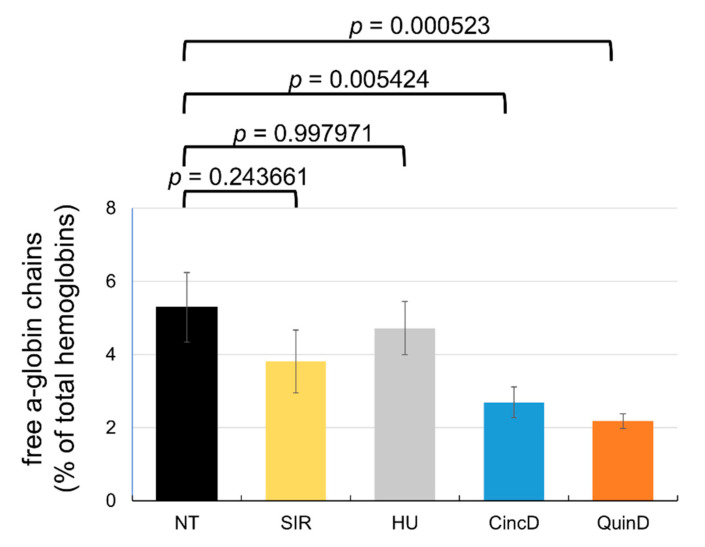
Effects of cinchonidine and quinidine on the free α-globin chains. ErPCs isolated from β-thalassemia patients were treated as indicated for 7 days with 100 nM sirolimus (SIR), 100 µM hydroxyurea (HU), 60 µM cinchonidine (CincD) and 30 µM quinidine (QuinD). The lysates were analyzed by HPLC (see Figure 5 for representative profiles) for quantification of free α-globin chains. The data represent the mean ± S.E.M. (*n* = 10).

## Data Availability

Most of the raw data are included in Appendix A. Additional information will be freely available upon request to the correspondence authors.

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
