# Peer review of "Treatment of Erythroid Precursor Cells from β-Thalassemia Patients with Cinchona Alkaloids: Induction of Fetal Hemoglobin Production"

_ijms, 2021, doi:10.3390/ijms222413433_

Round 1
Reviewer 1 Report
This manuscript describes the role of Cinchona alkaloids as an HbF inducing agent. The authors found that cinchona alkaloids (cinchonidine, quinidine and cinchonine) were natural HbF inducing agents in the K562 cell line and that cinchonidine and quinidine were able to induce HbF in erythroid progenitor cells isolated from β-thalassemia patients. In addition, they test the both compounds with an HbF inducer, sirolimus, and found synergic effects showing a maximal production of HbF.
Point 1: The introduction should be more extensive about cinchona alkaloids, and sirolimus. Why these molecules? Are they used as therapy and in which indication? Why using these molecules in b-thalassemias is a good idea? And so on. The last paragraph of the introduction is the first time Sirolimus was cited without presentation why the authors work with this molecule.
Point 2: In figure 2, the kinetic of sirolimus effect on K562 cells would be nice and permit to compare the kinetic of the cinchona alkaloids with the sirolimus one. Fig2 and Suppl data S1: Why are the X-axis different? The 2 graphs must be compared in terms to show a possible correlation between cell differentiation (benzidine positive cells%) and cell proliferation.
Point 3: In Figure 3, are the pictures in the same magnitude? The scale is lacking.
Point 4: In line 166: “On the contrary, when cinchonidine, quinidine and cinchonine were used in combination with 100 nM and 200 nM sirolimus, a sharp increase of the content of α-globin and γ-globin mRNAs was found (p < 0.01), fully in agreement with the effects of these compounds on sirolimus-induced K562 erythroid differentiation”. The “SHARP” increase is truth for α-globin mRNA but not for γ-globin mRNA. It is only an increase.
Point 5: In material and methods, the number of recruited patients is not indicated. Why are 9 patients in 2.4 and only 4 in 2.5? The patients are sous treatment(s) hydroxyurea or transfusion? How was the clinical circumstances of the recruitments; after a veino-occlusiv crises, at the diagnostic of the disease…? Were the sampling made when the patient have a stable disease?
Point 6: Fig 5 the Y-axis are presented with different scale, and therefore they are very complex to compare. The legend is not clear. Which chromatogram is for which patient? “ErPcs isolated from patient #5 (A-E) and patient #1 were treated with 100 nM sirolimus (B), 100 μM 198 hydroxyurea (C), 60 μM cinchonidine (D,G), 30 μM quinidine (E, H) for 7 days and the lysate ana-199 lyzed by HPLC.”
Point 7: The authors used the Hb chromatographic profil and calculated the % of the different types of Hb, 7 days after treatments, but what is about the concentration of Hb in cells. The % can increase but if the total concentration decrease the concentration of one types can increase in % but also decrease in concentration.
Point 8: Suppl Tab S1, S2 and S3 : There is no statistique in the table. Why? The total Hb concentration should be indicated and perhaps the concentrations of each type of Hb and free α-globin.
Point 9: The discussion should be more extensive. For example, the authors should discussed about: Which mechanism used by cinchona alkaloids and sirolimus, how to explain the synergic effects, why were cinchonidine, quinidine and cinchonine effects so different on Hb expression? Can these results be transposed to human treatments?
Author Response
REPLY TO REVIEWER 1
Dear Reviewer 1,
First of all, thank you very much for the comments to our paper “Treatment of erythroid precursor cells from β-thalassemia patients with Cinchona alkaloids: induction of fetal hemoglobin production”, by Cristina Zuccato, Lucia Carmela Cosenza, Matteo Zurlo, Ilaria Lampronti, Monica Borgatti, Roberto Gambari, and myself.
The paper was judged by the Editor to be interesting, but needing revision. We went through the reviewer comments and find all of them very useful for improving the scientific quality of the manuscript.
This is the list of the changes made answering to your points:
This manuscript describes the role of Cinchona alkaloids as an HbF inducing agent. The authors found that cinchona alkaloids (cinchonidine, quinidine and cinchonine) were natural HbF inducing agents in the K562 cell line and that cinchonidine and quinidine were able to induce HbF in erythroid progenitor cells isolated from β-thalassemia patients. In addition, they test both compounds with an HbF inducer, sirolimus, and found synergic effects showing a maximal production of HbF.
Comment. We thank the reviewer for her/his work and for the useful comments and suggestions.
Point 1: The introduction should be more extensive about cinchona alkaloids, and sirolimus. Why these molecules? Sirolimus: already described in the introduction section. Are they used as therapy and in which indication? Why using these molecules in b-thalassemias is a good idea? And so on. The last paragraph of the introduction is the first time Sirolimus was cited without presentation why the authors work with this molecule.
Answer. The issue raised is of key importance. The most important point is that both classes of compounds (Cinchona alkaloids and sirolimus (rapamycin) can be considered as repurposed drugs, therefore facilitating technological transfer and clinical applications. This is the key reason for designing the study. We confirm that they are used in therapy for other indications. In order to clarify these points, the following sentences have been added: “In addition, it should be underlined that molecules belonging to this family have been extensively used in therapy … with a facilitated strategy to reach technology transfer [29,34] (page 2, lines 73-76), “For instance, quinine, quinidine, cinchonidine and cinchonine alkaloids had a powerful bioimpact … it inhibits fibrillation [32-34]” (page 2, lines 76-78) and “Finally, it should be noted that sirolimus is a well-known drug, since it is employed for other therapeutic indications, such as kidney transplantation …. and a large variety of cancers [53-55] (pages 2 and 3, lines 97-101). As far as the fact that the use of these molecules for b-thalassemia might be a good idea, this is clear for sirolimus (and was explained in the introduction section, page 2, in sentences included in lines 84-97). As far as Cinchona alkaloids, the possible use in thalassemia was only hypothesized by the original interesting observation made by Iftikhar et al. [28]. Our study supports the concept that the activity of these molecules on thalassemic cells deserves attention.
Point 2: In Figure 2, the kinetic of sirolimus effect on K562 cells would be nice and permit to compare the kinetic of the cinchona alkaloids with the sirolimus one. Fig2 and Suppl data S1: Why are the X-axis different? The 2 graphs must be compared in terms to show a possible correlation between cell differentiation (benzidine positive cells%) and cell proliferation.
Answer. The effect of sirolimus is now shown in Supplementary Figure S3. A short comment has been added at page 4, lines 123-128. (“The induction of K562 erythroid differentiation by the studied Cinchona alkaloids is similar (with respect to the extent of the induced proportion of benzidine-positive cells) to that of …. are active at mM concentrations”). As far as the comparison of differentiation and cell growth, Supplementary Figure S2 has been added.
Point 3: In Figure 3, are the pictures in the same magnitude? The scale is lacking.
Answer. Thanks for raising this point. We included in the Figure bars for scaling. Magnitude of the pictures has been indicated.
Point 4: In line 166: “On the contrary, when cinchonidine, quinidine and cinchonine were used in combination with 100 nM and 200 nM sirolimus, a sharp increase of the content of α-globin and γ-globin mRNAs was found (p < 0.01), fully in agreement with the effects of these compounds on sirolimus-induced K562 erythroid differentiation”. The “SHARP” increase is truth for α-globin mRNA but not for γ-globin mRNA. It is only an increase.
Answer. We thank the reviewer for raising this point. The reviewer is right and therefore we emended the sentences as follows “a sharp increase of the content of α-globin mRNA (p < 0.01) and a less extensive but still significant (p < 0.05) increase of γ-globin mRNAs were found, fully in agreement…” (pages 6 and 7, lines 197-198).
Point 5: In material and methods, the number of recruited patients is not indicated. Why are 9 patients in 2.4 and only 4 in 2.5? The patients are sous treatment(s) hydroxyurea or transfusion? How was the clinical circumstances of the recruitments; after a veino-occlusiv crises, at the diagnostic of the disease…? Were the sampling made when the patient have a stable disease?
Answer. The number of recruited patients was ten. For all the patients the ErPCs were used. For all of them, the analysis was performed with singular treatment of HU, SIR, CincD and QuinD. ErPC cultures from five of them were also used for testing drug combinations. The patients were not under treatment(s) with hydroxyurea. All the patients were stable in their disease and transfusion depended. The patients were randomly selected, but we tried to have different genotypes, in order to confirm that the effects observed were not genotype-specific. This issue was considered by implementing the following sentence “In total, 10 patients were enrolled. Informed written consent … (this was done in ErPC cultures from five patients)” (page 7, lines 213-220). Chapter 4.1 (Patients recruitment) was implemented by adding the following sentence: “The recruited patients were all transfused and not under hydroxyurea treatment. Treatments were performed on ErPCs obtained from blood isolated from stable patients just before transfusion” (page 12, lines 395-398).
Point 6: Fig 5 the Y-axis are presented with different scale, and therefore they are very complex to compare. The legend is not clear. Which chromatogram is for which patient? “ErPcs isolated from patient #5 (A-E) and patient #1 were treated with 100 nM sirolimus (B), 100 μM hydroxyurea (C), 60 μM cinchonidine (D,G), 30 μM quinidine (E, H) for 7 days and the lysate analyzed by HPLC.”
Answer. In our mind Figure 5 was included to have a representative example of the analysis performed. We have in the new version of the Figure an HPLC profile in which the Y-axis of control versus CincD and QuinD treated samples are comparable.
Point 7: The authors used the Hb chromatographic profil and calculated the % of the different types of Hb, 7 days after treatments, but what is about the concentration of Hb in cells. The % can increase but if the total concentration decrease the concentration of one types can increase in % but also decrease in concentration.
Answer. One of the limits of our paper is that we used for this first proof-of-principle ErPCs, which are of interest since are primary erythroid cells, but are rather heterogeneous in their cellular composition. For this reason, in order to avoid any overinterpretation of the results we are just talking about increase in the % of HbF with respect to the total hemoglobin produced. We clarified this by adding the following short sentences: “As clearly evident, increase of the proportion of HbF (% of all the accumulated hemoglobins) was found …” (page 7, lines 232-233) and “Further studied employing analysis on the effects on transcriptome and proteome, as well as the confirmation of HbF increase …. is accompanied by a clinically relevant increase in the content of HbF in each treated erythroid cell” (page 11, lines 376-380).
Point 8: Suppl Tab S1, S2 and S3: There is no statistique in the table. Why? The total Hb concentration should be indicated and perhaps the concentrations of each type of Hb and free α-globin.
Answer. The relevant statistics have been included in Figure 7, which is commented in the text. We would like to underline that the trend is clear in the ErPC from the majority of the recruited patients. As expected, there is a lot of variability in respect to the starting levels of HbF and the response to the inducers. This was commented within the text.
Point 9: The discussion should be more extensive. For example, the authors should discussed about: Which mechanism used by cinchona alkaloids and sirolimus, how to explain the synergic effects, why were cinchonidine, quinidine and cinchonine effects so different on Hb expression? Can these results be transposed to human treatments?
Answer. The discussion section was implemented. We included a sentence clarifying the limit of our study. The sentence was “One of the limits of our study is that the mechanism(s) of action was not experimentally evaluated. Further studies are required to understand …. while sirolimus is firmly established as a mTOR inhibitor [66,67]” (page 11, lines 364-370). In a second sentence, we addressed what we think is important in order to hypothesis a possible application in human treatment. The sentence was “Further studied employing analysis on the effects on transcriptome and proteome, as well as the … as well as a careful analysis of the relationship between the effects on HbF and the presence of DNA polymorphisms associated with predisposition of the patients to high HbF induction” (page 11, lines 376-384). As far as possible impact on therapy and clinical relevance, our studies clearly indicate an effect of cinchonidine and quinidine on the reduction of the excess of a-globin chains. This relevant information was commented by including these new sentences: “Reduction of the excess of α-globin should be considered as an important objective .. determinant of the clinical severity of β-thalassemia [42]” (page 9, lines 296-299) and “Further studies will clarify whether the reduction of the excess of α-globin is associated with activation of autophagy, as elsewhere proposed [42]” (page 11, lines 354-356).
Dear Reviewer, thanks again for your comments.
Alessia Finotti
Reviewer 2 Report
Zuccato et al investigates “in vitro” the effect of three Cinchona alkaloids on fetal hemoglobin production in K563 cells and primary erythroid progenitor cell cultures from beta-thalassemic patients. Using benzamidine staining to analyze hemoglobinization, RT-qPCR to analyze gamma and alpha globin chain production, and HPLC to analyze HbF authors showed that cinchonidine, quinidine and to a lower extend cinchonine (tested only in K562 cells) induce fetal hemoglobin production and reduces free alpha globin chains. The paper is generally well written even if sometimes the long and complex phrases (“Italian style”) make difficult to get the concept. Even if the effect of the compounds is lower than expected there is a clear potential future development of pre-clinical studies using animal models. Moreover, it is of interest the synergistic effect with rapamycin/sirolimus. The study is descriptive and no possible mechanisms of action or activation/inhibition pathways are investigated, lowering the interest of the study.
Major points:
Could the lower cell growth of K562 (ie 40% reduction with CincD 60 uM, QuinD 40 uM and CincN 50 uM, which is consider a “suboptimal concentration”, at optimal concentration the reduction of cell growth) be due to a toxic effect? Was apoptosis measured? Please show also data using lower concentration of the compounds (ie 10 uM CincD and 20 uM QuinD) since it was described a non-antiproliferative effect of these agents at lower concentrations.
How do you explain the small increment of the expression of gamma globin in sirolimus and in CincD, QuinD treated K562 cells, much lower compared to previous reports? Iftikhar et al, 2019; Mischiati C ett al, BJH, 2004).
Minor points:
The induction of HbF in ErPC is very small in all of the conditions (closed to 1.25 fold maximum compared to control). Could authors comment? Is this increment clinically relevant?
Authors used sirolimus in combination with the Chincona tested alkaloids and not HU, which is the most used F-Hb inducer drug, widely used in sickle cell disease. Please comment and/or investigate the possible effect of these molecules in combination with HU.
It’s difficult to properly appreciate the induction of the gamma globin gene in most of the graphs/conditions represented in Fig 4, please use an appropriate scale for gamma globin expression (ie using a y-axis on the right). Please indicate the significance in the graphs.
Please specify details on how the compounds were administered to the ErPC cultures: were they added every day for 7 days or just when the medium was changed? How were the compounds dissolved?... Include this info in the mat and met section.
Please use dots instead commas in tables S1 and S3.
Please show benzidine staining of K562 of QuinD/SIR and CincN/SIR combinations in supplementary materials.
Fig. 8 please possibly show data from SIR at 100 nM, to uniform the graphs of ErPC treated cells.
Author Response
REPLY TO REVIEWER 2
Dear Reviewer 2,
First of all, thank you very much for the comments to our paper “Treatment of erythroid precursor cells from β-thalassemia patients with Cinchona alkaloids: induction of fetal hemoglobin production”, by Cristina Zuccato, Lucia Carmela Cosenza, Matteo Zurlo, Ilaria Lampronti, Monica Borgatti, Roberto Gambari, and myself.
The paper was judged by the Editor to be interesting, but needing revision. We went through the reviewer comments and find all of them very useful for improving the scientific quality of the manuscript.
This is the list of the changes made answering to your points:
Zuccato et al investigates “in vitro” the effect of three Cinchona alkaloids on fetal hemoglobin production in K563 cells and primary erythroid progenitor cell cultures from beta-thalassemic patients. Using benzamidine staining to analyze hemoglobinization, RT-qPCR to analyze gamma and alpha globin chain production, and HPLC to analyze HbF authors showed that cinchonidine, quinidine and to a lower extend cinchonine (tested only in K562 cells) induce fetal hemoglobin production and reduces free alpha globin chains. The paper is generally well written even if sometimes the long and complex phrases (“Italian style”) make difficult to get the concept. Even if the effect of the compounds is lower than expected there is a clear potential future development of pre-clinical studies using animal models. Moreover, it is of interest the synergistic effect with rapamycin/sirolimus. The study is descriptive and no possible mechanisms of action or activation/inhibition pathways are investigated, lowering the interest of the study.
Comment. We thank the reviewer for her/his comments and we hope to have acceptably modified the paper accordingly.
Major points:
Point 1. Could the lower cell growth of K562 (ie 40% reduction with CincD 60 uM, QuinD 40 uM and CincN 50 uM, which is consider a “suboptimal concentration”, at optimal concentration the reduction of cell growth) be due to a toxic effect? Was apoptosis measured? Please show also data using lower concentration of the compounds (ie 10 uM CincD and 20 uM QuinD) since it was described a non-antiproliferative effect of these agents at lower concentrations.
Answer. The antiproliferative effects are not unexpected, since several other HbF inducers exhibit this behavior. In this respect, we have not analyzed apoptosis. However, in all the experiments on ErPCs from the 10 recruited patients we determined the number of cells/ml (i.e. the proliferation rate) and we found that CincD and QuinD exhibit an effect very similar to SIR and HU. We included a comment on this at page 7, line 246-248 [The expected slight inhibitory effects of CincD and QuinD on ErPC cell proliferation were similar to those of the validated HbF inducers SIR and HU (Supplementary Figure S8)].
Point 2. How do you explain the small increment of the expression of gamma globin in sirolimus and in CincD, QuinD treated K562 cells, much lower compared to previous reports? Iftikhar et al, 2019; Mischiati C ett al, BJH, 2004).
Answer. As far as sirolimus, we employed concentrations (100 nM/200 nM) used for HbF induction in ErPCs (see Fibach et al., ref 37). The reason for this was to meet concentrations falling within the range of accumulation of sirolimus in vivo. As far as the effects of CincD and QuinD on K562 cells, the differences were probably on the use of different methodological approaches to detect benzidine-positive cells. It should be noted that the values of benzidine-positive cells of uninduced cells were very low (2-5% maximum) in our experiments, being higher in the research paper published by Iftikhar et al. Please, note however that the levels of gamma-globin mRNA were similar. In order to point out these issues the following short sentences have been included: “We used a concentration of sirolimus known to induce both K562 and erythroid precursor cells from b-thalassemia patients [37]” (page 12, lines 407-408); “In addition, the increase levels of γ-globin mRNAs were similar to those originally described by Iftikhar et al [28]” (page 6, lines 194-195); “Untreated K562 displayed a proportion of benzidine-positive cells which never exceeded 2-5% (see for instance the microphotographs presented in the left part of Figure 3)” (page 4, lines 150-151).
Minor points
Point 1. The induction of HbF in ErPC is very small in all of the conditions (closed to 1.25 folds maximum compared to control). Could authors comment? Is this increment clinically relevant?
Answer. This is a key issue that was not suitably commented in the previously submitted version of the paper. We agree on the fact that the increase of HbF in ErPCs was limited. This was the reason for verifying the activity of the molecules in ErPCs from 10 patients. In most of the cases to induction was similar (or higher) than the reference compound hydroxyurea and sirolimus, both of them used in clinical settings and demonstrated to retain therapeutic potential. On the other hand, our study is just a starting point on the road of validation of these treatments for a possible development of therapeutic protocols. More studies are necessary. To clarify this, several sentences were included in the discussion section. For instance, “In this respect it is interesting to observe that cinchonidine and quinidine might be more active that hydroxyureas, …. when clinical treatment of b-thalassemia and sickle-cell disease is considered” (page 11, lines 374-376) and “Further studied employing analysis on the effects on transcriptome and proteome, as well as the confirmation of HbF increase …… associated with predisposition of the patients to high HbF induction” (page 11, lines 376-384). As far as the induction of clinically relevant targets we have implemented the discussion by pointing out the very interesting effects on the excess of a-globin. To this aim we included the sentence “This reduction should be considered a key objective in the use of molecules for therapeutic interventions in the management …. short lifespan of the red-blood cells with associated ineffective erythropoiesis [42]” (page 11, lines 349-352).
Point 2. Authors used sirolimus in combination with the Chincona tested alkaloids and not HU, which is the most used F-Hb inducer drug, widely used in sickle cell disease. Please comment and/or investigate the possible effect of these molecules in combination with HU.
Answer. We decided to use combinations with sirolimus since this compound has been demonstrated to increase HbF in vivo and is employed at present in two clinical trials on beta-thalassemia. The reason for selecting sirolimus have been extensively discussed in the introduction. However, the possibility of testing the activity of Cinchona alkaloids in combination with hydroxyurea is highly relevant considering that HU might be considered one of the most important reference compounds in the treatment of thalassemia and sickle cell disease. This is now commented by including the sentence “Further studies will verify whether the Cinchona alkaloids employed in this study potentiate the activity of other HbF inducers, including hydroxyurea, extensively employed in the treatment of b-thalassemia and sickle-cell disease [62,63]” (page 11, lines 361-364).
Point 3. It’s difficult to properly appreciate the induction of the gamma globin gene in most of the graphs/conditions represented in Fig 4, please use an appropriate scale for gamma globin expression (ie using a y-axis on the right). Please indicate the significance in the graphs.
Answer. Thank you for raising this point. We have extensively modified the Figure and included relevant data on significance in the new version. All the statistical data are presented in Supplementary Figures S5 (concerning cinchonidine), S6 (concerning quinidine) and S7 (concerning cinchonine). The data presented are commented within slightly modified sentences.
Point 4. Please specify details on how the compounds were administered to the ErPC cultures: were they added every day for 7 days or just when the medium was changed? How were the compounds dissolved?... Include this info in the mat and met section.
Answer. The solubilization of the compounds is reported in Chapter 4.2. In order to answer to the point, the following short sentences were added “All the treatments were done by adding the compounds once at the beginning of the culturing period” (page 12, lines 409-410).
Point 5. Please use dots instead commas in tables S1 and S3.
Answer. Sorry for this. Tables have been emended.
Point 6. Please show benzidine staining of K562 of QuinD/SIR and CincN/SIR combinations in supplementary materials.
Answer. The required representative pictures have been included in Supplementary Fig.S4.
Point 7. Fig. 8 please possibly show data from SIR at 100 nM, to uniform the graphs of ErPC treated cells.
Answer. Yes, the concentration of sirolimus was 100 nM.
Dear Reviewer, thanks again for your comments,
Alessia Finotti
Round 2
Reviewer 1 Report
I thank the authors for the changes made in the article
Reviewer 2 Report
Authors adequately addressed all the issues.